# Preparation of TiO₂/Ag[BMIM]Cl Composites and Their Visible Light Photocatalytic Properties for the Degradation of Rhodamine B

**Xi Lin and Yanxia Li ***

Department of Chemical Engineering, Ocean College, Minjiang University, Fuzhou 350108, China; mjulinxi@mju.edu.cn

**\*** Correspondence: yxli@mju.edu.cn; Tel.: +86-13665034805

**Abstract:** In order to degrade toxic pollutants such as dyes during the process of sewage treatment, considerable attention has been paid to photocatalytic technologies. In this paper, $TiO_2/Ag[BMIM]Cl$ (1-butyl-3-methyl imidazolium chloride ([BMIM]Cl)) nanocomposites were prepared with $TiO_2$ as the carrier, silver ions as dopants and ionic liquids (IL) as modifiers. The morphologies, microstructures, crystalline structure and optical properties of the $TiO_2/Ag[BMIM]Cl$ nanospheres are investigated by transmission electron microscopy (TEM), X-ray diffraction (XRD), total organic carbon (TOC), and UV-vis diffuse reflectance spectrum (UV-vis DRS) techniques. The $TiO_2/Ag[BMIM]Cl$ nanocomposites can selectively degrade rhodamine B (Rh B) under visible light because of the unstable quaternary ammonium salt. The as-obtained nanocomposites exhibit better photocatalytic activity performance than pure $TiO_2$, $TiO_2/IL$, and $TiO_2/Ag^+$. The experimental results show that the Rh B degradation rate can reach 98.87% under optimized producing conditions by using the $TiO_2/Ag[BMIM]Cl$ composites as the catalyzer. It shows that simultaneous doping with silver ions and ionic liquids can significantly improve the photocatalytic activity of $TiO_2$ in Rh B degradation, indicating the formation of photosensitive AgCl in the process of $TiO_2/Ag[BMIM]Cl$ preparation. $Ag^+$ and IL addition exchange the band gap of $TiO_2$ and lengthen the visible wavelength range of the composite. The material has the advantages of low cost, facile preparation and reusability with the excellent degradation effect of Rh B.

**Keywords:** $TiO_2/Ag[BMIM]Cl$ composites; rhodamine B; visible photocatalytic degradation

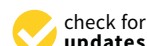



## 1. Introduction

The fast advance of industry has created many environmental problems. Our natural environment is suffering from unprecedented pollution. In recent years, environmental problems have attracted people's attention all over the world [1,2]. Semiconductor materials have a large potential to be used in the treatment of environmental sewage [3,4]. These pollutants contain dyes, farm chemicals, organic contaminants and so on, which can be degraded by photocatalytic reaction [5]. Photocatalysis using nanomaterials has emerged as a promising wastewater treatment process to overcome the major challenges faced by conventional technologies [6]. A large number of nanograins gather on the surface of nano materials to form a high concentration grain boundary, which promotes the photocatalytic effect [7].

Among the numerous nano materials, nano-TiO₂ has attracted considerable attention by virtue of its stable chemical property, high catalytic activity, popular price and so on [8,9]. Under photoinitiation, strong REDOX reactions can emerge on the surfaces of the nano photocatalyst, which can degrade the pollutants efficiently. TiO₂ has been widely applied in sewage treatment for its excellent photocatalytic properties [10]. However, due to its large band gap (3.0–3.2 eV) and low quantum effect, nano-TiO₂ can only be effectively excited by ultraviolet light, with poor adsorption capacity and low utilization under visible light.

This feature has hampered its feasible application in wastewater treatment [11]. Moreover, UV-light is very harmful to human skin.

Consequently, $TiO_2$ was modified through various strategies, including metals [12], non-metals [13,14], organisms [15,16] and ionic liquid [17] to decrease the energy gap and improve its visible light utilization and photocatalytic performance. Among them, silver-based nano/micro materials have received considerable attention because of their promising photosensitivity [12,18]. Silver-halide-included nanomaterials are wonderful self-sensitization photocatalysts for their intriguing electronic, optoelectronic, and photocatalytic properties [19]. Silver-halide with narrow bandgaps can break into $Ag^0$ with strong surface plasmon resonance effects and excellent electron storage property under light illumination [20].

Ionic liquids (IL) are composed of small volume inorganic anions ($BF_4^-$, $PF_6^-$) and large volume asymmetric organic cations, which exist in the form of ionic state at room temperature or adjacent temperature ($<100\ °C$) [21]. Compared with organic solvents, IL have lots of advantages, such as thermal stability, chemical stability, low melting point, good electrochemical properties, and it can dissolve various organic and inorganic materials [22]. Various researches on $TiO_2$-based IL with excellent photocatalytic activity have been reported [23]. It was already proved that the application of IL in the preparation of $TiO_2$ microparticles can improve the performance of $TiO_2$ under visible light irradiation [24]. IL were formed as supramolecular network of ions interacting via Coulombic forces, dipole−dipole and π–π interactions, hydrogen bonds, etc. These interactions, which also exist between IL and $TiO_2$, affect the surface architecture of $TiO_2$ nano/micro particles surrounded by the IL's protective layer, which in turn, changed the titania photo reactivity [25].

This study investigates the influence of visible light radiation on the composition of $TiO_2$ and Ag[BMIM]Cl and discusses the preparation conditions, including the crystal structure of $TiO_2$ and doping amounts of $Ag^+$ and IL. The formation and photocatalytic degradation mechanism of the $TiO_2$/Ag[BMIM]Cl composites were discussed. The prepared $TiO_2$-based photocatalysts exhibit outstanding photoactivity and reusability on the degradation of Rh B in water under visible light irradiation. This work provides a simple preparation method for the design and fabrication of novel functional photocatalysts.

## 2. Results and Discussion

### 2.1. Characterization of Catalysts

2.1.1. TEM Characterization

The TEM images of $TiO_2$ (A,B), $TiO_2$/$Ag^+$ (C,D), $TiO_2$/Ag[BMIM]Cl (E,F) are shown in Figure 1, respectively. The accumulated $TiO_2$ nanoparticles with a large number of mesopores can increase the specific surface area of $TiO_2$ nanoparticles, which has a positive effect on the degradation. There were no significant differences between the TEM images of $TiO_2$ (A,B) and $TiO_2$/$Ag^+$ (C,D) which indicated that silver is only doped in the form of $Ag^+$. However, Figure 1E,F clearly indicates the formation of spherical shape nanoparticles with diameters in the range of 10~30 nm, which is in consistent with the literature report [16]. This obvious difference comes from the newly formed AgCl. AgCl is a kind of photosensitive material, which can effectively improve the photocatalytic efficiency of $TiO_2$.

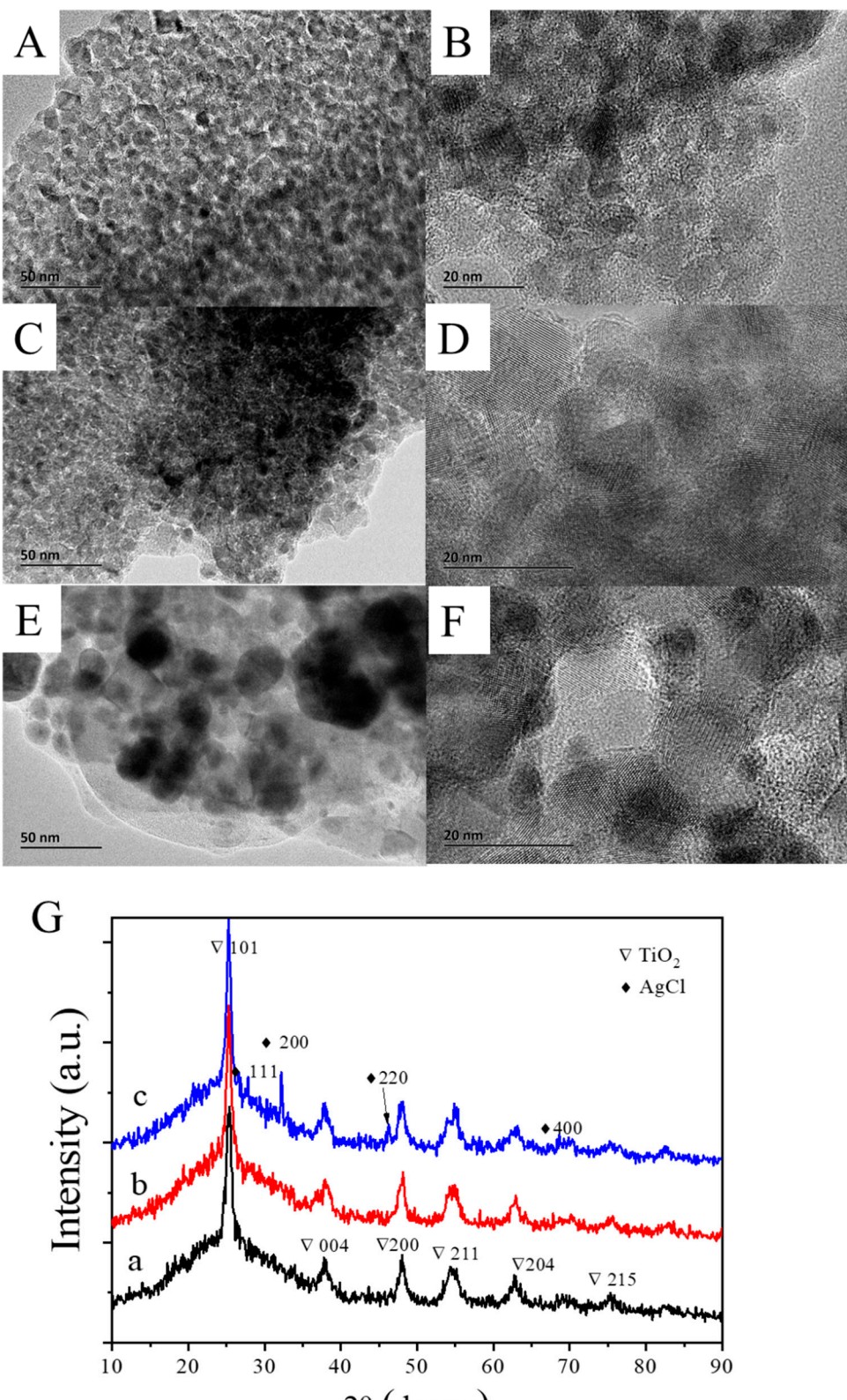

**Figure 1.** TEM images of $TiO_2$ (**A**,**B**), $TiO_2/Ag^+$ (**C**,**D**), $TiO_2/Ag[BMIM]Cl$ (**E**,**F**), and XRD patterns (**G**) of the $TiO_2$ (a), $TiO_2/Ag^+$ (b) and $TiO_2/Ag[BMIM]Cl$ (c).

#### 2.1.2. XRD Analysis

The XRD patterns of $TiO_2$ (a), $TiO_2/Ag^+$ (b) and $TiO_2/Ag[BMIM]Cl$ (c) are shown in Figure 1G. All of the three materials show diffraction peaks at 25.35, 27.75, 48.25, 55.15,

62.75 and 75.35°, which are respectively corresponding to the crystal plane (101) (004) (200) (211) (204) (215) of anatase (JCPDS No. 21–1272). This indicates that the prepared nano-$TiO_2$ is in the anatase phase with high purity. There is no new diffraction peak for $Ag^+$ doped on $TiO_2$ (curve b). As for $TiO_2$/Ag[BMIM]Cl, the diffraction peaks located at 26.15°, 32.07°, 46.37°, 68.57° are assigned to cubic AgCl (JCPDS no. 31-1238), which are indexed to the (111), (200), (220) and (400) crystal planes, respectively (curve c) [26]. These results indicate that $TiO_2$/Ag[BMIM]Cl composites include anatase $TiO_2$ and cubic AgCl.

### 2.1.3. Zeta Potential Characterization

The surface potential (Zeta potential) refers to the total potential difference between the surface of the particles and the solution in a dispersed phase. A high absolute value of surface potential shows strong electrostatic repulsion between particles indicating good dispersion and a stable system [27]. The zeta potentials of the $TiO_2$, $TiO_2$/$Ag^+$ and $TiO_2$/Ag[BMIM]Cl characterized by the Zeta potential analyzer (DT300, DTI, USA) were −11.41, −6.60, and 18.57 mV respectively. These data show that the $TiO_2$/Ag[BMIM]Cl composite is more stable because of the high absolute value of the Zeta potential ($|\zeta(TiO_2/Ag[BMIM]Cl)| > |\zeta(TiO_2)| > |\zeta(TiO_2/Ag^+)|$). The charge on the surface of $TiO_2$/$Ag^+$ reduces the stability of the materials. IL provide not only chloride ions to form AgCl, but also 1-butyl-3-methylimidazole group as ligands to improve the stability of nanomaterials.

### 2.2. Photocatalytic Degradation Properties of TiO₂/Ag[BMIM]Cl

Total organic carbon (TOC) is the total amount of organic matter in the solution and its variation reflects the degree of mineralization of the contaminants in the solution. Figure 2A shows photocatalytic degradation of TOC in Rh B solution with $TiO_2$, $TiO_2$/$Ag^+$ and $TiO_2$/Ag[BMIM]Cl composites for 2 h. However, $TiO_2$ and $TiO_2$/$Ag^+$ composite materials did not remove other organic compounds in the synthesis process. This shows that the TOC content of the solution with the $TiO_2$, $TiO_2$/$Ag^+$ composite is slightly higher than that of the original Rh B solution, and the photocatalytic degradation of the Rh B solution by the $TiO_2$/Ag[BMIM]Cl composite showed a relatively significant degree of TOC removal (removal rate 45.1%).

To investigate the photocatalytic effect of the visible light, the photocatalytic degradation experiments of the Rh B decolorization rates by $TiO_2$/Ag[BMIM]Cl under visible light, ultraviolet light and dark were carried out. A total of 20 mg $TiO_2$/Ag[BMIM]Cl was added in 5 mL aqueous solution of RhB (15 mg/mL). Incandescent lamp, ultraviolet lamp (254 nm + 365 nm) and darkness were selected as the irradiation sources. As shown in Figure 2B, the visible photocatalytic degradation efficiency is significantly higher than those under ultraviolet light and darkness catalysis after 2 h illumination. At the same time, it can also be seen from the decolorization rate under darkness conditions that the non-specific adsorption of $TiO_2$/Ag[BMIM]Cl is obviously low. This indicated that the high decolorization rate of $TiO_2$/Ag[BMIM]Cl mainly comes from visible light photocatalytic degradation.

The optical absorption of samples were presented by UV–vis DRS spectra. As shown in Figure 2C, the synthesized nanoparticles of $TiO_2$, $TiO_2$/$Ag^+$ and $TiO_2$/Ag[BMIM]Cl presented an obvious optical absorption signal in the region of optical wavelength ≤390 nm. There is still weak light absorption between 390 to 800 nm. Due to the continuous emission spectrum of incandescent lamp, it is conducive to the absorption of $TiO_2$/Ag[BMIM]Cl. Although there is strong light absorption of $TiO_2$/Ag[BMIM]Cl at 254 nm, the single wavelength UV lamp cannot provide enough energy of photocatalytic degradation only when the light intensity is increased. Without considering any type of transition (indirect or direct), the extrapolation yields Eg values of 2.96 eV for $TiO_2$ nanopowder, 2.97 eV for $TiO_2$/$Ag^+$ and 2.93 eV for $TiO_2$/Ag[BMIM]Cl [28]. The reason may be that the impurities of the Ag [BMIM] Cl are localized in the forbidden band of $TiO_2$. Therefore, the excitation

photon energy needed for electron transition from impurity to conduction band is reduced, which is conducive to photocatalytic degradation.

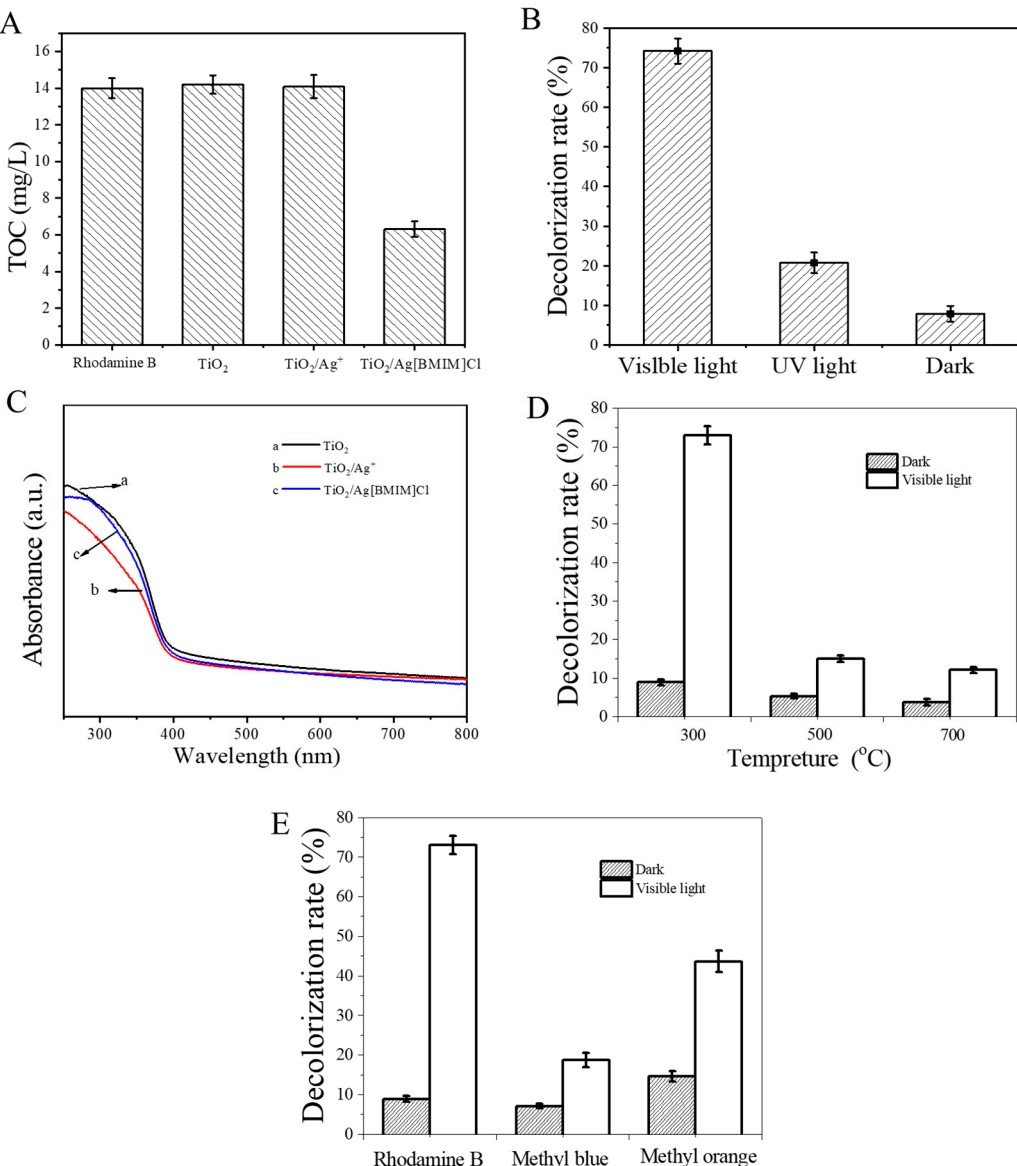

**Figure 2.** TOC analysis (**A**) of the $TiO_2$, $TiO_2/Ag^+$ and $TiO_2/Ag[BMIM]Cl$ after visible photocatalytic degradation for 2 h; decolorization rates of $TiO_2/Ag[BMIM]Cl$ after photocatalytic degradation for 2 h (**B**); Rh B under different irradiation sources; UV–vis DRS spectra of $TiO_2$, $TiO_2/Ag^+$ and $TiO_2/Ag[BMIM]Cl$ (**C**) Rh B catalyzed by $TiO_2/Ag[BMIM]Cl$ with different preparation temperatures of $TiO_2$ under visible light (**D**); different dyes under visible light (**E**).

It is known that calcination temperature can obviously influence the microstructure and photocatalytic activity of $TiO_2$ materials [29]. As shown in Figure 2D, with the increase of calcination temperature of $TiO_2$ from 300 to 500 °C, the visible photocatalytic degradation efficiency of $TiO_2/Ag[BMIM]Cl$ reduced obviously. This phenomenon is in consistent with the literature [30]. As the calcination temperature increases, the crystal type of $TiO_2$ gradually changes from anatase to rutile. Compared with anatase, rutile has a more intense absorbance and a smaller bandgap [31]. Therefore, anatase can degrade the pollutants better than rutile. The lower temperature is favorable for the formation of anatase. As expected, mesoporous $TiO_2$ nanoparticles calcined at 500 and 700 °C show virtually no

photoactivity, as they contain the rutile phase. In order to remove the reactants during $TiO_2$ preparation, 300 °C for 4 h was chosen as the optimized condition.

To verify the degradation effect of Rh B by the prepared $TiO_2$/Ag[BMIM]Cl materials, the dyes with three different structures were compared. Figure 2E shows the decolorization rates of $TiO_2$/Ag[BMIM]Cl after visible photocatalytic degradation for 2 h. From Figure 2E, it can be seen that the prepared composite has the largest degradation efficiency of Rh B, followed by methyl orange and finally, methyl blue. This result shows that the prepared composite has a degradation priority for the degradation of the relevant dyes. From the structural formula of the three dyes (Table 1), it can be seen that Rh B contains unstable quaternary ammonium salt. The ethyl group bonded by quaternary amine is easy to be decomposed under visible-light photocatalytic degradation, followed by tertiary amine. The proposed photocatalytic degradation pathway of Rh B can be referred to the reference [32]. The tertiary amine in methyl orange is also easily degraded. However, the sulfonic acid group with electron withdrawing group is attached to the para position of diazonium on the benzene ring forming a stable structure. The three sulfonic groups in methyl blue make the nitrogenous group more stable, so it is not easy to be degraded under visible light.

**Table 1.** Physicochemical characteristics of the three dyes.

| Dye | Chemical Formula | $M_w$ (g/mol) | $\lambda_{max}$ (nm) |
|---|---|---|---|
| Rh B |  | 479.01 | 566 |
| Methyl blue |  | 799.80 | 662 |
| Methyl orange |  | 327.33 | 505 |

The visible photocatalytic degradation properties of the as-synthesized samples were further investigated. The $TiO_2$, $TiO_2$/IL, $TiO_2$/Ag$^+$, and $TiO_2$/Ag[BMIM]Cl were added into 5 mL 15 mg/mL Rh B aqueous solution under visible-light irradiation for 2 h. As can be seen from Figure 3A, all these materials can degrade Rh B to different extent. The degradation rates of Rh B in $TiO_2$, $TiO_2$/IL, $TiO_2$/Ag$^+$ and $TiO_2$/Ag[BMIM]Cl were 2.71%, 41.11%, 49.11% and 72.97%, respectively (Figure 3B). This result showed that the $TiO_2$/Ag[BMIM]Cl composites has the best degradation effect on Rh B.

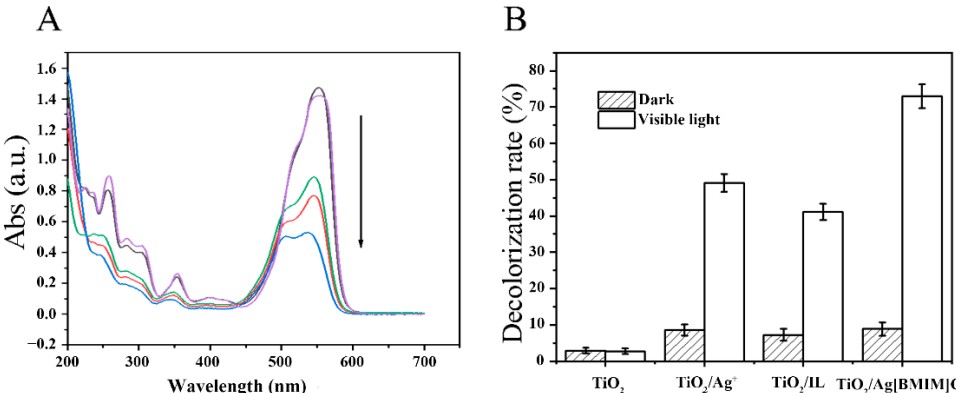

**Figure 3.** Ultraviolet spectra of Rhodamine B (**A**) and decolorization rates of different nanomaterials (**B**) in the preparation process of the $TiO_2/Ag[BMIM]Cl$ after photocatalytic degradation for 2 h. From top to bottom: Rhodamine B (15 mg/L), $TiO_2$, $TiO_2/IL$, $TiO_2/Ag^+$, $TiO_2/Ag[BMIM]Cl$.

## 2.3. Catalytic Effect of $Ag^+$ and IL Dosages during the $TiO_2/Ag[BMIM]Cl$ Processing

It can be seen from Figure 4A that the decolorization rate first increased with $Ag^+$ dosage and reached the best when the molar ratio of $Ag^+$ and $Ti^{4+}$ [$n(Ag^+)/n(Ti^{4+})$] was 10%, and then decreased rapidly. The possible reason is that excessive $Ag^+$ adsorbed on the surface of the $TiO_2/Ag[BMIM]Cl$ composite had a bad effect on the catalytic performance of the composite. From the XRD patterns (Figure 4B), the diffraction peaks of AgCl at about 32.07° and 46.37° became visible when the molar ratio of $Ag^+$ and $Ti^{4+}$ reached 10%. This result was in consistent with Figure 4A which indicating that the cubic structure of AgCl formed in the process of $TiO_2/Ag[BMIM]Cl$ preparation played a key role in the visible light photocatalytic degradation. On the other side, the decolorization rate changed with IL dosage. When the molar ratio of IL and $Ti^{4+}$ [$n(IL)/n(Ti^{4+})$] reached 1%, the degradation efficiency of the $TiO_2/Ag[BMIM]Cl$ on Rh B achieved was the best, and then decreased gradually (Figure 4C). The XRD patterns (Figure 4D) further explained that the formation of AgCl played an important role in the degradation process. Excessive IL dissociate on the surface of the material and interfere with the photocatalytic degradation of the $TiO_2/Ag[BMIM]Cl$ composites.

## 2.4. Evaluation of Photocatalytic Degradation Effects

### 2.4.1. Effect of pH

The experimental results from the photocatalytic experiments were further analyzed in order to define the treatment conditions to maximize the degradation efficiency of $TiO_2/Ag[BMIM]Cl$. Visible-light induced degradation of Rh B in different pH solutions with $TiO_2/Ag[BMIM]Cl$ nanocomposites was studied. Figure 5A showed that the initial pH values had an obvious effect on the photodegradation efficiency of Rh B. As it can be seen from Figure 6A, alkaline solution is not conducive to photocatalytic degradation. There was almost no adsorption of Rh B at high pH values due to the function of negatively charged –COO$^-$ [33]. From the surface electronic characteristics of the catalyst, low pH can improve photocatalytic activity. However, in terms of the surface adsorption performance of the catalyst and the existing form of Rh B, the low pH value of the solution is not conducive to the improvement of the photocatalytic activity. This is mainly because Rh B exists in the form of cation in acidic solution. The lower the pH value of the solution, the more positive charges on the surface of the catalyst. Due to the electrostatic repulsion of the same charge, it is not conducive to the adsorption of Rh B on the surface of the $TiO_2/Ag[BMIM]Cl$, which resulted in the reduced photocatalytic activity. Under the influence of the above two opposite factors, the highest catalytic activity of the composites does not appear at the lowest pH value. It can be concluded that the best degradation effect was achieved when pH was 5.

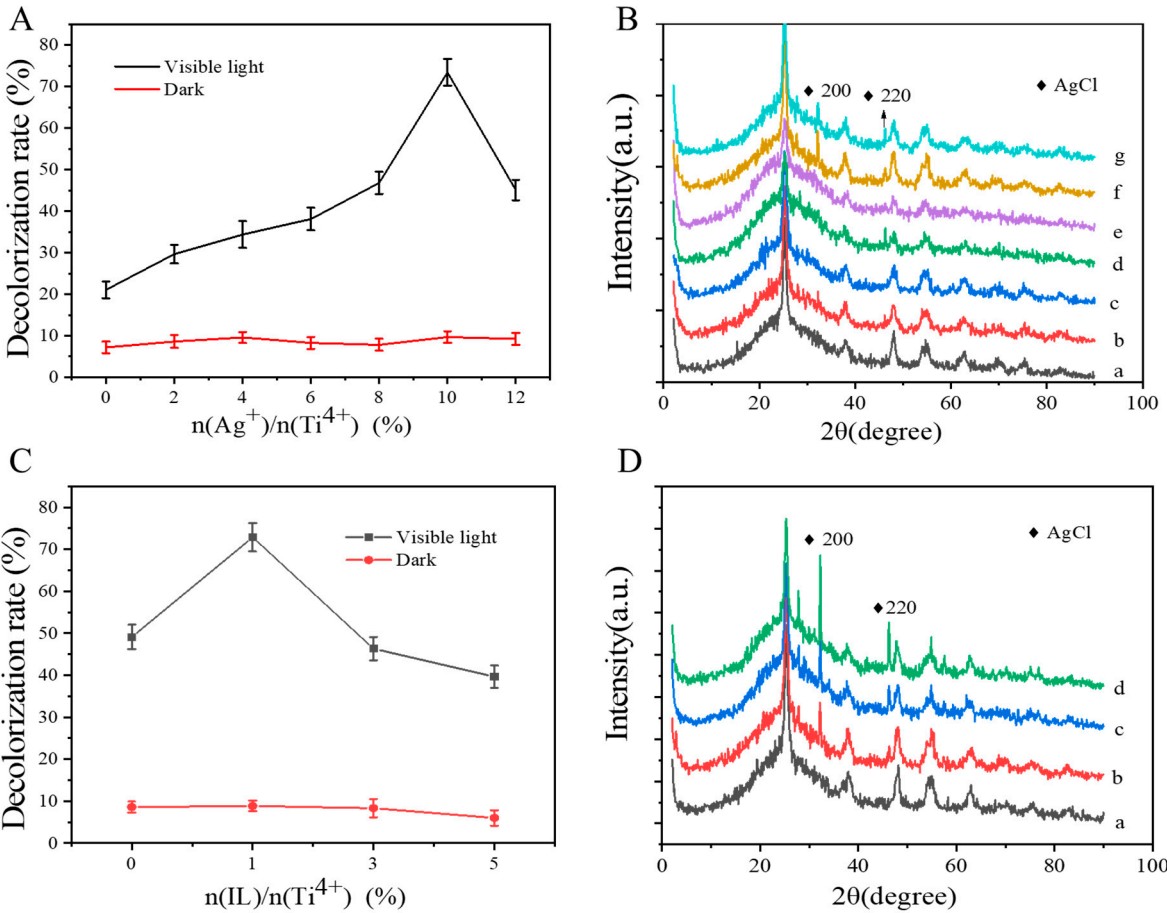

**Figure 4.** Visible light photocatalytic effect of $Ag^+$ (**A**) and ionic liquid (**C**) dosages during the $TiO_2/Ag[BMIM]Cl$ processing, XRD patterns of the $TiO_2/Ag[BMIM]Cl$ prepared at different amounts ((**B**) from a to g $n(Ag^+)/n(Ti^{4+})$ at 0%, 2%, 4%, 6%, 8%, 10%, 12%, IL: 1%; (**D**): from a to d $n(IL)/n(Ti^{4+})$ at 0%, 1%, 3%, 5%, $Ag^+$: 10%).

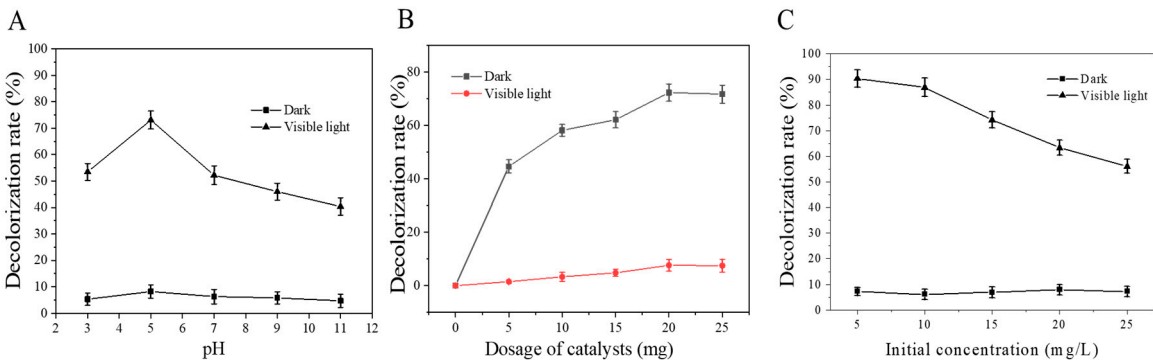

**Figure 5.** Visible light photocatalytic effect of the $TiO_2/Ag[BMIM]Cl$ changed with pH (**A**), dosage (**B**) and initial Rh B concentration (**C**).

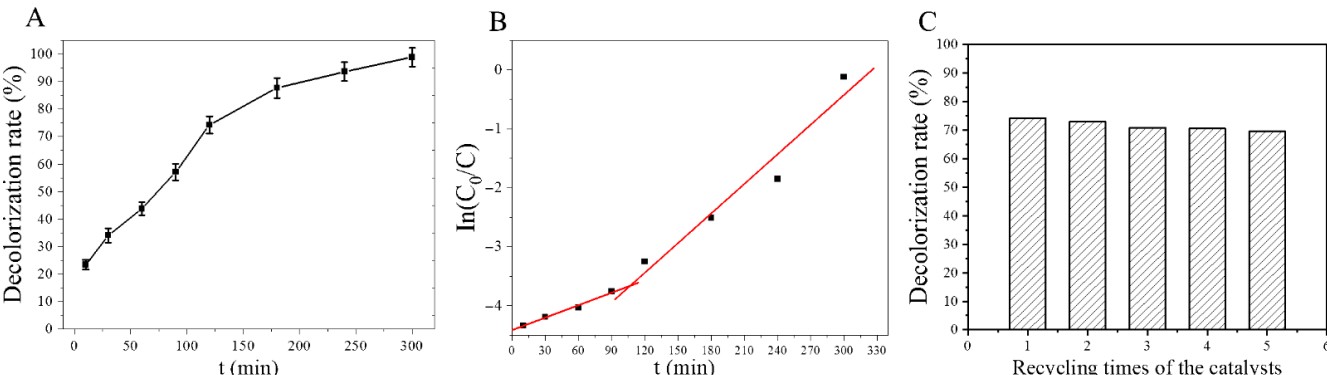

**Figure 6.** Visible light photocatalytic effect of the TiO$_2$/Ag[BMIM]Cl changed with illumination time (**A**), the first−order−kinetic plots for TiO$_2$/Ag[BMIM]Cl (**B**) and reusability performance (**C**).

### 2.4.2. Effect of Catalyst Dosage

The effect of catalyst dosage on the degradation of Rh B was investigated by using different adsorbent dosage over the range of 0~25 mg, and the decolorization rate was analyzed after 2 h illumination over 15 mg/mL Rh B. Figure 5B showed that the degradation efficiency increased with the increase of catalyst dosage. This could be attributed to the increased surface area of TiO$_2$/Ag[BMIM]Cl and the availability of interaction with Rh B. The decolorization rate got a peek value at 20 mg. Therefore, 20 mg catalyst was selected as the optimized dosage.

### 2.4.3. Effect of Degradation on Initial Rh B Concentration

The effect of different initial Rh B concentration on the photocatalytic performance over TiO$_2$/Ag[BMIM]Cl at loading 20 mg was evaluated under visible light irradiation for 2 h. Figure 5C shows that the decolorization rate of Rh B reduced with the increase of its initial concentration gradually. It is the significant amount of visible light absorbed by Rh B molecules rather than TiO$_2$/Ag[BMIM]Cl at high Rh B concentrations that resulting in the reduction of photocatalytic activity [33]. The decolorization rate could still be 56.07% when the initial Rh B concentration was 25 mg/L. This indicates that the prepared catalyst still shows good catalytic activity at high concentration of Rh B.

### 2.4.4. Kinetics of Photocatalytic Reaction

Kinetic study on visible light photocatalytic Rh B degradation by the TiO$_2$/Ag[BMIM]Cl was evaluated. As shown in Figure 6A, with the increase of illumination time, the decolorization rate increased gradually. The decolorization rate reaches 98.87% under visible light irradiation for 300 min indicating that the prepared catalyst has high catalytic activity and reaction rate.

The kinetics of photocatalytic degradation of Rh B can be depicted by the first-order equation:

$$In\left(\frac{C_0}{C}\right) = kt \tag{1}$$

where $C_0$ and $C$ are the initial concentration of Rh B before degradation and after illumination. The pseudo-first-order rate constant $k$ (min$^{-1}$) can be estimated from the linear fitting of $In(C_0/C)$ versus irradiation time ($t$). The plots of $In(C_0/C)$ vs. $t$ for TiO$_2$/Ag[BMIM]Cl can be observed in Figure 6B. Two distinct slopes are found at the dividing line of 100 min indicating that the degradation process is divided into two stages. By the fitting of $In(C_0/C)$ versus $t$, the $k_1$ values (before 100 min) and $k_2$ values (after 100 min) for the photocatalytic degradation reactions of Rh B over TiO$_2$/Ag[BMIM]Cl can be estimated and listed in Table 2. Table 2 shows that the pseudo-first-order rate constant $k_2$ after 100 min is significantly higher than $k_1$ before 100 min, indicating the reaction rate increases after 100 min.

This may be for the reason that the amount of visible light is also absorbed by Rh B. With the extension of illumination time, the concentration of Rh B decreased accordingly. As a result, the increasing amount of visible light absorbed by photocatalyst leads to an increase in photocatalytic activity.

**Table 2.** Linear relationships of $In(C_0/C)$ vs. $t$ based on pseudo-first-order model.

| Linear Range (min) | k (min$^{-1}$) | R$^2$ |
|---|---|---|
| 0~100 | 0.00704 | 0.9885 |
| 100~300 | 0.01674 | 0.9422 |

2.4.5. Stability and Reusability of a Photocatalyst

The stability and reusability of a photocatalyst are important in practical applications. Therefore, the stability of the photocatalyst was further tested and the cyclic photocatalytic experiment was carried out. It can be seen from Figure 6C that after 5 recycles, the photocatalyst did not exhibit any significant loss of activity (more than 90% was kept), indicating its high stability and reusability during photocatalytic process for practical applications.

*2.5. Mechanism of Photocatalytic Degradation*

The structural formula of 1-butyl-3-methyl imidazolium chloride ([BMIM]Cl, $C_8H_{15}ClN_2$) is shown in Figure 7.

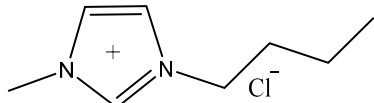

**Figure 7.** The chemical structure of IL.

Based on the previous reports, and the above results, the mechanism of photocatalytic degradation is proposed. The energy belt of $TiO_2$ consisted of an empty conduction band and a valence band filling with electrons. The forbidden band forms between conduction band (CB) and valence band (VB). At the beginning of illumination, the surface of $TiO_2$ material produced holes ($h^+$) and photogenerated electrons ($e^-$) by excitation. The holes can transfer fleetly to the surfaces of Ag[BMIM]Cl and the unsuccessfully recombined electrons transfer to the surfaces of $TiO_2$. Electrons on the surface of $TiO_2$ stars a chain reaction with $O_2$ in the solution to produce $O_2^-$ and other reactive oxygen species (ROS).

The photosensitivity of Ag[BMIM]Cl can effectively absorb visible light which promotes the combination of $Cl^-$ and photogenerated hole to form strong oxidizing $Cl^0$ radical, which are able to oxidize Rh B and then be reduced to $Cl^-$. The $Cl^0$ radical can be used to oxidize the organic contaminant, due to its strong oxidation activity. Furthermore, the ligand (BMIM) from IL modified the surface of the composites effectively to promote the electron capture of Ag ions on semiconductor surface. This modification not only restrained the recombination of electrons and holes, but also promoted the synthesis of active groups. In this way, photocatalysts get higher surface active sites and adsorptive properties, thus improving the photocatalytic degradation property of composites.

Through the analysis of the reaction products, the possible mechanism for the photocatalytic process is proposed as follows:

$$TiO_2 + Ag^+ \rightarrow TiO_2/Ag^+ \tag{2}$$

$$TiO_2/Ag^+ + [BMIM]Cl \rightarrow TiO_2/Ag[BMIM]Cl \tag{3}$$

$$TiO_2/Ag[BMIM]Cl + hv \rightarrow TiO_2/Ag^*[BMIM]Cl \tag{4}$$

$$TiO_2/Ag^*[BMIM]Cl \rightarrow TiO_2/Ag[BMIM]Cl^-(h^+) + e^- \tag{5}$$

$$h^+ + H_2O \rightarrow H^+ + \bullet OH \tag{6}$$

$$O_2 + e^- \rightarrow \bullet O_2{}^- \tag{7}$$

$$TiO_2/Ag[BMIM]Cl^{\cdot}(h^+) \rightarrow TiO_2/Ag[BMIM]^+ + Cl^{\mathbf{0}} \tag{8}$$

$$Cl^0 + Rh\ B \rightarrow degradation\ products + Cl^- + CO_2 + H_2O \tag{9}$$

$$\bullet O_2{}^- + Rh\ B \rightarrow degradation\ products + O_2 + CO_2 + H_2O \tag{10}$$

$$\bullet OH + Rh\ B \rightarrow degradation\ products + CO_2 + H_2O \tag{11}$$

$$TiO_2/Ag[BMIM]^+ + Cl^- \rightarrow TiO_2/Ag[BMIM]Cl \tag{12}$$

## 3. Materials and Methods

### 3.1. Materials

Tetrabutyl titanate $[CH_3(CH_2)_3O]_4Ti$, silver nitrate ($AgNO_3$), rhodamine B (Rh B), methylene blue, methyl orange, acetic acid, anhydrous ethanol of analytical grade were received from Xinyuhua (Fuzhou, China). 1-butyl -3-methyl imidazolium chloride ([BMIM]Cl) (IL) was purchased from Aladdin Co. Ltd. (Shanghai, China). Milli-Q (ULUPURE, Merck Millipore, Kenilworth, NJ, USA) purified water was used for all experiments described here. All experiments were performed in triplicates. The error bars showed the standard deviation (SD) in all figures.

### 3.2. Preparation of Mesoporous TiO$_2$

The $TiO_2$ nanoparticles were synthesized by sol–gel method [34]. Firstly, glacial acetic acid (2.5 mL) and anhydrous ethanol (5 mL) were added into 5 mL deionized water under magnetic stirring for 10 min (Solution A). Tetrabutyl titanate (5 mL) was added into 20 mL anhydrous ethanol with vigorous stirring for 10 min (Solution B). Fresh solution A and B were simultaneously prepared. Then, in a typical preparation, Solution B was slowly poured into solution A to obtain a milky transparent sol under intense stirring. The colloid was left for several hours, dried in an oven (60 °C) at low temperature to obtain yellow crystal. The crystal was ground into powder with a mortar, placed in a muffle furnace, and calcined at 300 °C for 4 h to obtain nano-$TiO_2$.

### 3.3. Synthesis of TiO$_2$/Ag[BMIM]Cl

The obtained nano-$TiO_2$ was ground into powder. 1.0 g $TiO_2$ powder was dispersed in deionized water followed by dissolving 0.214 g $AgNO_3$ under intense stirring. An aqueous solution (20 mL) contained 0.02 g IL was added into the $TiO_2$ dispersion. The mixture was stirred at room temperature for 45 min. After centrifugation, the sediments were washed several times with ethanol and deionized water, and dried at 60 °C overnight.

### 3.4. Photocatalytic Degradation of Rh B

A total of 20 mg $TiO_2$/Ag[BMIM]Cl nanomaterials were added into 5 mL Rh B aqueous solution. A 30 W incandescent lamp was used as the irradiation source. The degradation of the dyes was investigated using UV–vis spectrophotometer (UV-2250 Shimadzu, Kyoto, Japan) at 566 nm. The decolorization rate (η) was calculated by the change of absorbance before and after photocatalytic degradation.

$$\eta = \frac{A_0 - A}{A_0} \times 100\% \tag{13}$$

where $A_0$ is the initial absorbance of the dye before degradation and $A$ is the absorbance of the dye after illumination.

## 4. Conclusions

A novel catalyst of $TiO_2$/Ag[BMIM]Cl was successfully prepared. The photocatalytic performances were evaluated based on the photocatalytic degradation of Rh B solution

under visible light. It showed that the doping of Ag[BMIM]Cl could significantly improve the visible photocatalytic activity of $TiO_2$ in Rh B selective degradation. Compared with the ultraviolet, visible light has higher catalytic efficiency. The $TiO_2$/Ag[BMIM]Cl catalyst displays the highest photocatalytic activity (98.87% decolorization rate of Rh B after visible light irradiation for 300 min under the optimized preparation and degradation reaction conditions). Moreover, the composites exhibited good recycle reusability. The $TiO_2$/Ag[BMIM]Cl catalyst with excellent photoactivity has a large potential for use in degrading organic pollutants under visible light.

**Author Contributions:** Data curation, X.L.; Methodology, X.L. and Y.L.; Project Administration, Y.L.; Writing—Original Draft, Y.L.; Writing—Review and Editing, Y.L. and X.L. All authors have read and agreed to the published version of the manuscript.

**Funding:** This research was funded by NSFC (21405075), Fujian province natural science foundation (2020J01835), the opening foundation of Fujian Key Laboratory of functional marine sensing materials, Minjiang University (MJUKF-FMSM201914).

**Conflicts of Interest:** The authors declare no conflict of interest.

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
