# Peer review of "Preparation of TiO2/Ag[BMIM]Cl Composites and Their Visible Light Photocatalytic Properties for the Degradation of Rhodamine B"

_catalysts, doi:10.3390/catal11060661_

Round 1
Reviewer 1 Report
The paper is interesting and the topic is suitable for the Coatings journal. However, in my opion the introduction need to be improved, by adding recent important studies on the same topis. See for example Progress in Organic Coatings, , ,
Author Response
Answer: Thanks for the reviewer’s remind. The references were added to the introduction according to the reviewer’s suggestion. The place which had been modified was underlined in red.

Reviewer 2 Report
Reviewer Report
Manuscript Number: catalysts -1177891
Title: Preparation of TiO2/Ag[BMIM]Cl composites and their visible light photocatalytic properties for the degradation of Rhodamine B
The paper deals with the obtaining, by a facile liquid reaction, and characterization of TiO2/Ag[BMIM]Cl composites materials as potential photocatalysts in wastewater treatment. Even if the authors report a lot of interesting results, demonstrating the high photoactivity and reusability of TiO2/Ag[BMIM]Cl composite in the degradation of rhodamine B (Rh B) under visible light irradiation, the paper is not very well written especially related to the results explanation in a logical, clear flow, easy to follow by a reader. Moreover, some of results are not sufficiently explained and/or correlated, and the form, in term of writing and language, of the manuscript need to be improved. Based on those mentioned above, I consider that this manuscript is not acceptable for publication in Catalysts in view of journal impact factor and standards.
Here I list few arguments that motivate my decision:
The abstract is too general, half of it referring to the well-known properties of the TiO2 photocatalyst, therefore the abstract may be reformulated in order to highlight the novelty of the researches supported by significant results obtained for (optimized) TiO2/Ag[BMIM]Cl composite in the degradation of rhodamine B (Rh B) under visible light irradiation.
- In Introduction:
The authors repeat he same information about TiO2, as semiconductor photocatalytic material, during two sentences representing 6 lines of text (lines 36 – 41).
- In Results and discussion:
I did not understand why the authors chose to start with the description of the photodegradation mechanism and not with the materials characterization (crystalline phases composition, surface morphology and potential, absorbance spectra), the evaluation of the photocatalytic process (the influences of different parameters on the photodegradation of Rh B under Vis light irradiation etc.) and then the formulation of the mechanism, based on the correlation of obtained results.
Although many interesting results have been obtained, which demonstrates an important work done, their reporting is still chaotic, without conclusive explanations and correlations.
For example, one of the studied materials is TiO2/Ag+, TiO2 doped with Ag+ ions, but none of the experimental investigations clearly showed the doping of TiO2 with Ag+; also, the presence of AgCl is confirmed only by X-ray diffraction, not by other analysis techniques (EDS, SAED, HRTEM, XPS).
The degradation mechanism of RhB in the presence of TiO2/Ag[BMIM]Cl catalyst is a general one, based on literature data, in which important data are missing, such as the energy bandgap of TiO2/Ag[BMIM]Cl and the values of CB and VB. The role of AgCl on the photodegradation mechanism is not clearly discussed.
Other general observations:
- English language must be revised by a native English speaker;
- The authors are sometimes using repetitive sentences;
- A large number of bibliographic references (31) are given in the paper, including almost half (15) are recent (2019-2020) and 3 are from current year.
Author Response
- The abstract is too general, half of it referring to the well-known properties of the TiO2 photocatalyst, therefore the abstract may be reformulated in order to highlight the novelty of the researches supported by significant results obtained for (optimized) TiO2/Ag[BMIM]Cl composite in the degradation of rhodamine B (Rh B) under visible light irradiation.
Answer: Thank you for point out that. We have revised the abstract for highlighting the novelty of the researches, and now it was improved. Please refer to lines 11-16, 20-21. The place which had been modified was underlined in red.
- In Introduction:
The authors repeat the same information about TiO2, as semiconductor photocatalytic material, during two sentences representing 6 lines of text (lines 36 – 41).
Answer: Thank you for point out that. Through our careful examination, we introduced TiO2 in the abstract which repeat the same information about TiO2 in introduction. Therefore, the information about TiO2 in abstract (Titanium dioxide, as a photocatalyst with stable chemical property, strong corrosion resistance, non-toxicity and low cost, has attracted numerous attentions in recent years. However, the catalytic ability of titanium dioxide is poor, and the photogenerated electrons and holes are inclined to combine together very soon inside the catalyst and on the surfaces of catalyst, which also contributes to the poor performance of TiO2.) was deleted to preserve the integrity of the introduction and avoid the repetition of the content.
- In Results and discussion:
I did not understand why the authors chose to start with the description of the photodegradation mechanism and not with the materials characterization (crystalline phases composition, surface morphology and potential, absorbance spectra), the evaluation of the photocatalytic process (the influences of different parameters on the photodegradation of Rh B under Vis light irradiation etc.) and then the formulation of the mechanism, based on the correlation of obtained results.
Answer: Thank you for point out that. We are very sorry that there is no reasonable layout in the framework of the paper. The part of “Mechanism of photocatalytic degradation” was moved to the last part of the result and discussion according to the reviewer’s advisement. And now it was improved.
- Although many interesting results have been obtained, which demonstrates an important work done, their reporting is still chaotic, without conclusive explanations and correlations.
For example, one of the studied materials is TiO2/Ag+, TiO2 doped with Ag+ ions, but none of the experimental investigations clearly showed the doping of TiO2 with Ag+; also, the presence of AgCl is confirmed only by X-ray diffraction, not by other analysis techniques (EDS, SAED, HRTEM, XPS).
Answer: Thank you for point out that. Several characterization methods including TEM, XRD, Zeta potential, TOC and UV–vis DRS spectroscopy were employed to analyze TiO2, TiO2/Ag+ and TiO2/Ag[BMIM]Cl. In fact, we have to admit that many characterization methods are difficult to confirm the structure. For TiO2/Ag+ nanocomposite, there was no difference by characterization of TEM, XRD, TOC and UV–vis DRS compared to TiO2 indicating no new nanoparticles formation. The surface of TiO2 prepared by sol gel method contains a certain amount of hydroxyl, which is easy to bond with Ag+ in solution. This is confirmed by the change of Zeta potential. The whole Ag+ and IL doping process was carried out in neutral aqueous solution at room temperature under no reductant. Padervand confirmed the formation of AgBr in the synthesis of the Ag[1-butyl 3-methyl imidazolium]Br nanospheres(Chemical Engineering Communications, 203:1532–1537, 2016). XRD The results of XRD analysis indicate that TiO2/Ag[BMIM]Cl composites include anatase TiO2 and cubic AgCl. No diffraction peak of Ag was found. Therefore, We confirmed the structs of TiO2/Ag+ and TiO2/Ag[BMIM]Cl from three aspects: literature background, characterization results and theoretical derivation.
The degradation mechanism of RhB in the presence of TiO2/Ag[BMIM]Cl catalyst is a general one, based on literature data, in which important data are missing, such as the energy bandgap of TiO2/Ag[BMIM]Cl and the values of CB and VB. The role of AgCl on the photodegradation mechanism is not clearly discussed.
Answer: Thank you for point out that. According to the UV–vis DRS spectra, the energy band was further discussed. Please refer to lines 130-135. At the same time, the role of AgCl on the photodegradation mechanism was also further discussed. The primary roles of AgCl are to enhance the absorption of visible light with the of property photosensitivity and provide Cl0 radical. Please refer to lines 272-274. And the modified parts were highlighted in red.
- Other general observations:
- English language must be revised by a native English speaker;
Answer: We had revised the manuscript carefully according to the reviewer’s comments. And now the manuscript was improved. The place which had been modified was underlined in red.
- The authors are sometimes using repetitive sentences;
Answer: We had revised the manuscript carefully according to the reviewer’s comments, in particular, the parts of abstract and introduction. Now the manuscript was improved. The place which had been modified was underlined in red.
- A large number of bibliographic references (31) are given in the paper, including almost half (15) are recent (2019-2020) and 3 are from current year.
Answer: we’re sorry that we ignored this problem. In the process of writing the introduction, we tried to select the latest literatures, but in the experimental research and theoretical discussion, we referred to some early literatures. Three literatures were added for the improvement of the research background.

Reviewer 3 Report
This paper reports the preparation of TiO2/Ag composites and their visible light photocatalytic properties. The experiments appear to have been very carefully performed, and the author’s explanation seems reasonable.
Thus, this might be acceptable for Catalysts. Nevertheless, the author should make a few modifications/additions, as indicated at the end of this text.
- p2, line 75, “1- butyl -3- methyl” should be changed into “1-butyl-3-methyl”. Please delete the space (three points).
2. p2. line 86, What is “Cl0”? Does it mean Cl radical (Cl atom)? I don’t understand it, please give us more your explanation.
Author Response
- p2, line 75, “1- butyl -3- methyl” should be changed into “1-butyl-3-methyl”. Please delete the space (three points).
Answer: Thank you for point out that. The “1-butyl-3-methyl” was revised.
2. p2. line 86, What is “Cl0”? Does it mean Cl radical (Cl atom)? I don’t understand it, please give us more your explanation.
Answer: Thank you for point out that. As the reviewer’s remind, the Cl0 means Cl radical. For better expression, we further explain this concept. Please refer to lines 272-274. And the modified parts were highlighted in red.

Round 2
Reviewer 2 Report
Manuscript Number: catalysts -1177891
Title: Preparation of TiO2/Ag[BMIM]Cl composites and their visible light photocatalytic properties for the degradation of Rhodamine B
The paper deals with the obtaining, by a facile liquid reaction, and characterization of TiO2/Ag[BMIM]Cl composites materials as potential photocatalysts in wastewater treatment. The authors report a lot of interesting results, demonstrating the high photoactivity and reusability of TiO2/Ag[BMIM]Cl composite in the degradation of rhodamine B (Rh B) under visible light irradiation. The revised form of the manuscript is significantly improved. However, there are still issues that need to be revised before being accepted for publication.
The issues I am referring to are the following:
In Abstract:
Line 10 – the abbreviation IL must be entered in the text: “ionic liquids (IL)”;
Must be specified in the text what BMIM means.
- In Introduction:
In lines 36–37 and 39-40, the authors repeat the same properties for TiO2: “stable chemical property, high catalytic activity, popular price” and “high catalytic activity, chemical stability and low cost”; please reformulate/change the text so that the same properties are not repeated;
At the end of line 58 – must be completed with “light”: visible light irradiation;
Lines 66-67, the text “The composite technique of TiO2 and Ag[BMIM]Cl effectively improved the visible light photocatalytic efficiency” is unclear, so please reformulate or delete it.
In Results and discussion:
Line 84 is XRD analysis not analyzation;
The characterization of the photocatalytic materials includes TEM, XRD and Zeta potential analyses, so I suggest to the authors to discuss first these analyses and then those related to photocatalytic process, including here the TOC results and discussions;
The results for TOC analysis should be listed in Figure 3, not in the same Figure with XRD patterns;
Lines 115-116 must be reformulated “To verify investigate the photocatalytic effect of visible light on/of….. WHAT??, photocatalytic degradation experiments under visible light, ultraviolet light and dark were carried out";
Lines 116-117 “20 mg TiO2/Ag[BMIM]Cl was added into 5 mL Rh B aqueous solution” – something is missing here and is “in 5 mL aqueous solution of RhB (15 mg/mL)”;
Line 120 - pay attention to the manuscript writing (this is available for entire manuscript): after 2h illumination.
Line 126 - “wavelength ≤ 390 nm”;
Line 136 – “It is known to us that…”;
Lines 212 – 214 should be modified as: “The effect of catalyst dosage on the degradation of Rh B (15mg/mL) was investigated by agitating with using different adsorbent dosage over the range of 0-25 mg, and the decolorization rate was analyzed after 2 h of illumination.”;
Line 261 (263) – I suggest to change Scheme with Figure, for the formula of C8H15ClN2;
Line 280-281 “Through the analysis of the reaction products, the possible (or proposed, not both) mechanism for the reaction photocatalytic process is as follows: “, and start with (1) in reactions numbering, not with (2).
Lines 312-313 “The degradation of the dyes was determined investigated by using a UV–vis spectrophotometer ..”
Author Response
Dear Reviewer:
Thanks for the reviewer’s remind. We revised a point-by-point response to the reviewer’s comments. Please see the attachment of revised manuscript. Please see the attachment.
